# DRAW IT LIKE EUCLID: TEACHING TRANSFORMER MODELS TO GENERATE CAD PROFILES USING RULER AND COMPASS CONSTRUCTION STEPS

## ABSTRACT

We introduce a new method of generating Computer Aided Design (CAD) profiles via a sequence of simple geometric constructions including curve offsetting, rotations and intersections. These sequences start with geometry provided by a designer and build up the points and curves of the final profile step by step. We demonstrate that adding construction steps between the designer's input geometry and the final profile improves generation quality in a similar way to the introduction of a chain of thought in language models. Similar to the constraints in a parametric CAD model, the construction sequences reduce the degrees of freedom in the modeled shape to a small set of parameter values which can be adjusted by the designer, allowing parametric editing with the constructed geometry evaluated to floating point precision. In addition we show that applying reinforcement learning to the construction sequences gives further improvements over a wide range of metrics, including some which were not explicitly optimized.

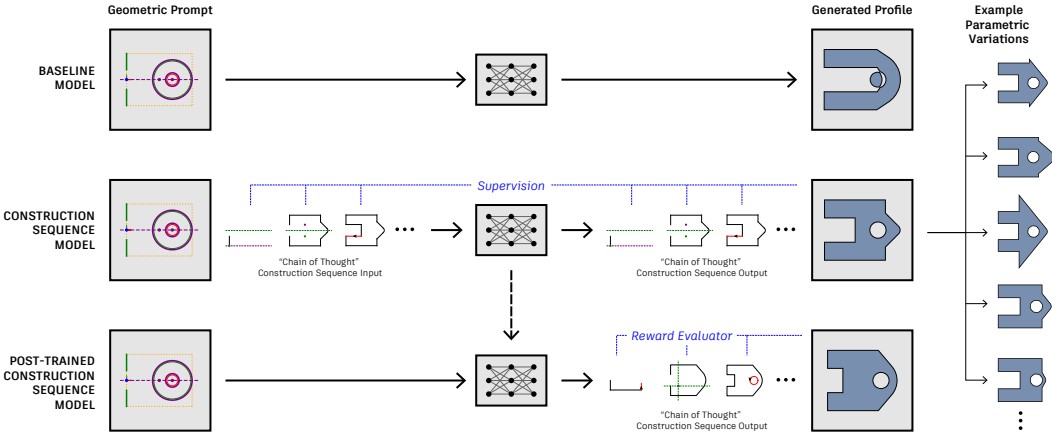

Figure 1: Our model (middle) generates CAD profiles through a sequence of ruler, compass, and protractor constructions, starting from the designer's initial "prompt" geometry (left) and building step by step toward the final profile. Our approach generates profiles that more accurately match the designer's input and contain fewer self-intersections, compared to a baseline model (top) which omits construction steps and maps directly from geometric prompt to profile. We further refine the construction sequence model during post-training (bottom), where rewards help guide the generation of valid construction sequences. As the sequences encode parametric relationships, families of related profiles (right) can be created from a single construction trace.

## 1 INTRODUCTION

Computer Aided Design (CAD) tools play a key role in shaping nearly all manufactured objects. CAD models are created by specifying collections of lines, arcs and circles which enclose 2d regions

called profiles. These can then be extruded or revolved to define solid volumes, which can be combined using boolean operations to build complex shapes. Geometric constraints can be added to the profiles, enforcing relationships between the curves. For example, lines can be constrained to be parallel, circles concentric and curves can be forced to meet tangentially. The distances and angles between specific curves can also be controlled using parameters, which can be adjusted to modify the shape while maintaining critical aspects of the design like symmetries, regular patterns and constant thicknesses.

While machine learning models for the generation of 2d CAD geometry have shown great advances in recent years, these methods have two main limitations. Firstly, all current methods suffer from limited accuracy. Methods which create geometry as a sequence of discrete tokens representing quantized points or coordinates Seff et al. (2020); Willis et al. (2021); Ganin et al. (2021); Para et al. (2021); Wu et al. (2023) have their precision limited by the spacing of the quantization grid, while diffusion methods Xu et al. (2024); Fan et al. (2024); Lee et al. (2025) are limited by the convergence of the diffusion process and the decoding of latent vectors. Secondly, while some methods can predict constraints and dimensions, this is done as a second step after the geometry has been generated. An external constraint solver is then required to post-process the curve geometry and produce the final shape. It has been shown that applying generated constraints in post-process can frequently move the geometry Para et al. (2021); Ganin et al. (2021) which is undesirable when the changes significantly alter the design Casey et al. (2025).

In this work we investigate an alternative strategy for generating 2d profiles, using a unified sequence which defines both the geometry and shape properties which would usually be enforced by constraints. Inspired by recent work in language models, which show that an intermediate chain of thought (CoT) can greatly improve the accuracy of the final output Wei et al. (2022); Kojima et al. (2022), we wonder whether the generation of some intermediate geometry might assist with CAD generations tasks. We notice that when CAD designers build up shapes, they often start by sketching some intermediate "construction geometry" which defines important aspects of a design like symmetry lines or construction circles. Constraints are then added between the construction geometry and profile curves to enforce properties like coincidence and symmetries. Inside the geometric constraint solver, the graph formed by the geometry and constraints is recursively processed to yield a fine grained sequence of simple geometric constructions which builds up the final shape Owen (1991); Bouma et al. (1995). These construction sequences play a role similar to the algorithm execution traces which have been employed to great effect in a variety of search problems Yang et al. (2022); Lehnert et al. (2024); Gandhi et al. (2024).

In this paper we conduct experiments training transformer models on intermediate construction sequences of ruler, compass and protractor construction steps similar to those used in geometric constraint solvers. As the high level task of shape generation is broken down into small atomic steps with closed form solutions, the performance of the network at solving these subtasks and combining them to build a consistent "CAD program" can be measured separately. The shapes can be controlled using "prompt geometry" provided at the start of the sequence and used as inputs to subsequent constructions. Once generated, the sequences can be replayed with floating point precision, allowing accurate values known at inference time to be propagated through the constructions to build the final profile. The generated geometry is parameterized using a small number values, which can be varied when the sequences are replayed allowing parametric edits to the shape as shown in the right of Figure 1.

We demonstrate quantitatively that the introduction of construction sequences improves the performance of the generative models, reducing self-intersections and proving enhanced adherence to the design requirements. In addition, we show that applying reinforcement learning over the entire construction sequence can further improve results as shown in the language modeling case Shao et al. (2024); DeepSeek-AI et al. (2025).

Our contributions can be summarized as follows

- We introduce a domain specific language which builds 2d profile geometry as a sequence of ruler, compass and protractor constructions steps. The construction steps can be replayed with floating point accuracy, allowing parametric editing of the shape.

- We show that geometry generated with sequences which include these intermediate construction steps have fewer self-intersections, superior accuracy when auto-completing partial designs and better adherence to other design requirements like symmetry lines.

- We show that reinforcement learning, with reward functions which discourage self-intersecting geometry, leads to improvement over a wide range of metrics, many of which are not explicitly optimized.

## 2 RELATED WORK

### 2.1 ALGORITHM TRACES

A number of works have explored training neural networks to mimic the logical steps conducted by heuristic algorithms. Vinyals et al. (2015) showed that RNNs could replicate some basic geometric algorithms like finding the convex hull and building a delaunay triangulation. Yang et al. (2022) experimented with monte carlo tree search traces for maze navigation, robotic manipulation and Atari games. Lehnert et al. (2024) used traces from $A^*$ search to learn to solve mazes and Gandhi et al. (2024) showed how search and backtracking capabilities could be used to play the game Countdown. In this work we investigate the applicability of these techniques to CAD, utilizing algorithm traces similar to those used in geometric constraint solvers.

### 2.2 SKETCH GENERATION

The availability of large scale constrained sketch datasets Seff et al. (2020); Ganin et al. (2021) opened the task of CAD sketch generation and sketch auto-constraining to the community. Seff et al. (2020) presented an autoregressive model based on message passing networks which generated parametric sketches by iteratively predicted the edges and node attributes of the constraint graph. An external geometric constraint solver was then required to construct the final geometry and the complexity of the resulting sketches was limited. Willis et al. (2021) showed that unconditional generation of 2d sketches was possible by first generating a list of points and then using PointerNetworks Vinyals et al. (2015) to group these to define curves. Ganin et al. (2021), Para et al. (2021) and Seff et al. (2021) presented networks which generated 2d curves and then, in a second step, used PointerNetworks to predict constraints and dimensions conditioned on this geometry. Because these architectures predict design intent only after the geometry has been generated, they cannot leverage the constraints to guide the curve placement. Instead, an external constraint solver must be applied as a post-process. As shown in Ganin et al. (2021), Para et al. (2021) and Casey et al. (2025), this can shift the positions of sketch geometry, revealing that the initial curve placement did not reflect the intended design. In contrast, the construction sequence representation introduced here enables design intent to be predicted before geometry is constructed, allowing the network to incorporate it directly when placing points and curves.

### 2.3 CHAIN OF THOUGHT IN CAD

Khan et al. (2024) used parametric recipes from the DeepCAD dataset Wu et al. (2021) to define a natural language description of the CAD modeling features used to created the solids. Guan et al. (2025) used Deepseek-V3 to convert these descriptions into a natural language CoT followed by CadQuery code. A Qwen2.5-7B-Instruct model was fine tuned on this data and the GRPO reinforcement learning algorithm with a chamfer distance based reward function was used to further enhance results. Li et al. (2025) used DeepSeek-R1 to generate a natural language CoT and CAD commands from a text description. The CoT was passed to Gemini-2.0 along with images of the generated CAD model to provide visual feedback in an iterative refinement loop. The natural language CoTs employed by these models were used an an auxiliary representation along side the executable code. While they includes statements related to design intent, these are not defined in a formal language which can be directly executed by CAD kernels.

### 2.4 REINFORCEMENT LEARNING FOR CAD

A number of recent papers have shown promising results applying reinforcement learning to a variety of CAD related tasks. Casey et al. (2025) fine tuned a CAD the auto-constrainer model from

Seff et al. (2021) with a number of RL algorithms. This was shown to improve the fraction of entities fully defined by the constraints while reducing the fraction of sketches where geometry moved when the constraints were applied. Yin et al. (2025) studied the recovery of a parametric feature recipe from B-Rep models. An Actor-Critic network selected faces of the target B-Rep to extrude or revolve and rewards were based on the similarity between the recovered and target shapes. Chen et al. (2025) used Direct Preference Optimization (DPO) to improve the performance of an image to CAD command sequence network and Kolodiazhnyi et al. (2025) studied the use of reinforcement learning to improve CAD reconstruction from point clouds.

# 3 DATA

## 3.1 DATASET CREATION

Our training data is derived from the CAD models in the ABC dataset Koch et al. (2019). The Open Cascade (2025) modeling kernel is used to create the profiles, by slicing each B-Rep solid with 5 equally spaced section planes with normals along each of the three coordinate system axes. Disjoint regions are separated so that each extracted profile has one outer loop and zero or more inner loops. This results in closed profile loops consisting of line segments, arcs and circles. The data deduplicated procedure converted each profile into a graph with nodes as the vertices and curves as edges. The edges were labeled with the curve type and nodes were labeled using the vertex coordinates, quantized into 8x8 bins. The Weisfeiler Lehman graph hash Shervashidze et al. (2011) was then computed and profiles with duplicated hashes were removed. The data was then split into 95% train, 3% validation and 2% test.

The extraction of the construction sequences from the raw profile geometry utilizes a set of simple heuristic algorithms which are described briefly here. A detailed description of each of the algorithm's phases can then be found in Appendix B. To simulate the input of a designer, we start our sequences with information used for shape control which we refer to as a "geometric prompt". Rather than auto-completing a sketch from a random subset of sketch geometry as in Seff et al. (2021), we extract line segments from the convex hull and the positions of internal circular loops. The area and bounding box of the profile, along with any symmetry lines are also included. Next in an "analysis phase", we identify geometric relationships between curves such as parallel lines, concentric circles and fillet arcs. These relationships are translated into construction steps, such as curve offsetting and filleting operations. We then built a bipartite dataflow graph in which nodes represent geometry and construction steps and directed edges represent how geometry flows into and out of the operations. Initially this graph will contain cycles and redundant branches, which are removed in a "graph simplification phase". The construction sequence is then obtained using a lexicographical topological sort, in which the order of the curves in the final profile is used to resolve ambiguities in the topological sort order. The data extraction process yielded a total of 318,208 profiles with corresponding construction sequences.

## 3.2 LEARNED SEQUENCES

Our experiments utilize sequences with the following three components. The sequences start with the "geometric prompt" which is used to control the shape. Next we include the construction steps, which act like a chain of thought, starting with the prompt geometry and constructing the points and curves required to define the final profile. Finally we have the profile geometry, which is analogous to the final answer returned by a reasoning LLM.

The construction steps represent simple geometric operations such as curve offsetting, curve-curve intersections, curve reversal, mirroring points over symmetry lines and the construction of fillet arcs. A few examples of supported construction steps are provided in Table 1, and the remainder are listed in Table 6 of Appendix A.1. Each construction step has an operation type, a list of input geometry and a list of output geometry. The output geometry of one step can be utilized as the input to subsequent steps, building up a description of the dataflow graph. The construction steps are ordered such that the curves are created in the order they appear in the profile, with the first curve chosen so that its end point is closest to the bottom left hand corner. Details of the domain specific language (DSL) and tokenization used to encode the sequences are in Appendix A.

Table 1: Ruler and compass construction steps. Examples of input geometries to the construction steps are shown in blue and their output in red.

| Description | Explanation | Example |
|---|---|---|
| `CircleOffsetCircle` **Input:** `circle₁, offset` **Output:** `circle₂` | Given an oriented circle and a positive offset distance, find and return the offset circle. | |
| `LineXLine` **Input:** `line₁, line₂` **Output:** `point` | Given two lines, find and return their intersection point. | |
| `LineOffsetLine` **Input:** `line₁, offset` **Output:** `line₂` | Given a directed line and an offset distance, find and return the line offset from this line to the left hand side by the offset distance. | |
| `LineXCicle` **Input:** `line, circle` **Output:** `point(s)` | Given a line and a circle, find and return the intersection point(s). | |

## 4 METHOD

### 4.1 SUPERVISED LEARNING

We train an autoregressive decoder only transformer on the sequences described in 3.2 with a cross entropy loss. In our experiments we use 8 heads, a depth of 8, and embedding dimension of 1024, and an attention head dimension of 128. We use the Adam optimizer Kingma & Ba (2014) with a learning rate of $3e - 4$ and dropout of 0.1. Training was performed on 4 RTX 6000 GPUs.

Two variants of the model were trained. A baseline model which includes only the information in the geometric prompt and then the geometry of the final profile, and a construction sequence model which additionally includes the intermediate construction steps. A comparison of the performance of these models is given in Section 5.2

### 4.2 REINFORCEMENT LEARNING (RL)

The fine-tuning of the profile generation model can be formulated as follows; the profile generation model $\pi_\theta(\tau|x)$ generates a profile sequence $\tau$ for geometric prompt $x$. Given a set of geometric prompts $D = \{x_i\}_i^N$ and reward function $r$ that provides a scalar value $r(x, \tau)$ which evaluates the quality of a profile sequence $\tau$ and how well it satisfies the geometric constraints prescribed by $x$. RL finetuning proceeds by maximizing the expected rewards

$$\max_\theta \mathbb{E}_{x \sim \rho} \mathbb{E}_{\tau \sim \pi_\theta(\cdot|x)}[r(x, \tau)], \tag{1}$$

where $\rho$ represents the distribution of the geometric prompts, and $\theta \in \mathbb{R}^d$ denotes for the training parameters of the profile generation model.

### 4.2.1 REWARD DESIGN

While reinforcement learning from human feedback (RLHF) tends to involve vague and subjective human preferences, the task of CAD profile generation allows for direct and objective evaluation of

the generated profile sequences through measurable validity metrics, which eliminates the need for a learned reward model. The rewards used for RL are defined as follows:

- Reward for syntactically valid profile sequence $\tau$:

  $r_{\text{no self-intersection}}$: fraction of profiles without self-intersecting curves,

  $r_{\text{no short edges}}$: fraction of profiles without edges shorter than a predefined minimum length,

- Reward for syntactically invalid profile sequence:

  $r_{\text{invalid profile}}$: penalty for generated sequences that produce syntactically invalid DSL code and cannot be detokenized.

### 4.2.2 POLICY GRADIENT METHODS

We focus on three sequence-level policy gradient methods: ReMax (Li et al., 2024), GRPO (Shao et al., 2024), and RLOO (Ahmadian et al., 2024). These methods compute policy gradients using the total log-probability of the generated sequence and apply a REINFORCE-style estimator with learned baselines (Weaver & Tao, 2001). In contrast to PPO (Schulman et al., 2017), which performs token-level policy updates using individual log-probabilities at each decoding step, these approaches operate at the sequence level and eliminates the need for a separately trained value network. We apply them to fine-tune the profile generation policy.

**ReMax** (Li et al., 2024) uses the reward of a sequence generated by greedily decoding from the policy network as a baseline to normalize the rewards of sequences sampled stochastically.

With sequences $\tau$ sampled from policy $\pi_\theta(\tau|x)$, the ReMax baseline $b_{\theta,\text{ReMax}}$ is $\text{argmax}(\pi_\theta(\tau|x))$, and the policy gradient objective for ReMax is:

$$\mathbb{E}_{\tau \sim \pi_\theta(\cdot|x)}\Big[\big(r(x,\tau) - b_{\theta,\text{ReMax}}\big) \cdot \nabla_\theta \log \pi_\theta(\tau \mid x)\Big] \tag{2}$$

**Group Relative Policy Optimization (GRPO)** (Shao et al., 2024) samples a group of $G$ individual profile sequences $\{\tau\}_{g=1}^{G}$ for every geometric prompt $x$. The advantage of the $g$-th profile sequence is calculated by normalizing the group-level rewards $\{r(x,\tau_g)\}_{g=1}^{G}$:

$$A_g = \frac{r(x,\tau_g) - \text{mean}(\{r(x,\tau_g)\}_{g=1}^{G})}{\text{std}(\{r(x,\tau_g)\}_{g=1}^{G})}. \tag{3}$$

The GRPO policy gradient objective is:

$$\mathbb{E}_{\{\tau_g\}_{g=1}^{G} \sim \pi_\theta(\cdot|x)}\Big[\min(\psi_g A_g, \text{clip}(\psi_g, 1-\epsilon, 1+\epsilon)A_g) - \beta D_{\text{KL}}(\pi_\theta \| \pi_{\text{ref}})\Big], \tag{4}$$

where:

$$\psi_g = \frac{\nabla_\theta \pi_\theta(\tau_g \mid x)}{\pi_{\text{ref}}(\tau_g \mid x)}, \qquad D_{\text{KL}}\big(\pi_\theta \| \pi_{\text{ref}}\big) = \frac{1}{\psi_g} + \log\psi_g - 1 \tag{5}$$

**REINFORCE-Leave-One-Out (RLOO)** (Ahmadian et al., 2024) samples a group of $G$ individual profile sequences $\{\tau\}_{g=1}^{G}$ for every geometric prompt $x$. The reward for each sample within a group $r_g$ serves all other samples as a baseline, resulting in the policy gradient objective as follows:

$$\mathbb{E}_{\{\tau_g\}_{g=1}^{G} \sim \pi_\theta(\cdot|x)}\Big[\frac{1}{G}\sum_{g=1}^{G}[(r_g - \frac{1}{G-1}\sum_{i \neq g} r_i) \cdot \nabla_\theta \log \pi_\theta(\tau_g \mid x)]\Big]. \tag{6}$$

# 5 RESULTS

## 5.1 EVALUATION METRICS

Evaluation metrics for 2d parametric profile generation can be broadly divided into two categories: *validity metrics*, which assess whether a generated profile sequence is syntactically correct and geometrically sound, and *prompt satisfaction metrics*, which evaluate how well the generated profile shape adheres to the constraints and properties specified in the input geometric prompt.

### 5.1.1 VALIDITY METRICS

Validity metrics measure whether the generated profiles conform to both the syntactic requirements of the DSL and the implicit geometric expectations of a well-formed profile shape. These metrics include:

**Syntactic validity**: Whether a generated sequence can be successfully detokenized under the strict syntactic rules of the DSL.

**No self-intersection**: Whether the resulting profile is free of self-intersections.

**No short edges**: Whether all edges exceed the minimum length defined by the quantization bin size.

These are typically reported as boolean indicators, aggregated as the overall fraction of valid profile generations.

### 5.1.2 PROMPT SATISFACTION METRICS

The degree to which a generated profile adheres to the geometric prompt can then be quantitatively evaluated, enabling direct and objective assessment of prompt satisfaction. Some of the prompt satisfaction metrics include:

**Area**: measured by the difference between the area prescribed in the geometric prompt and that of the generated profile.

**Line segments**: for each line segment specified in the prompt, the metric is computed based on the presence of profile line segments that are collinear, overlapping, and equal in length. The distance between the end points of the requested and generated line segments is also recorded.

**Center-of-gravity**: measured by the distance between the center of gravity defined in the geometric prompt and that of the generated profile.

**Holes**: measured by the distances between the hole centers defined in the geometric prompt and that of the generated profile.

**Symmetry lines**: measured by the intersection over union (IoU) of the profile and its reflection across the symmetry line, averaged over all symmetry lines in the prompts.

**Outer bounding box**: measured by the intersection over union between the outer bounding box defined in the geometric prompt and that of the generated profile.

**Fraction of tangent continuous vertices**: measured by the difference between the fraction of tangent continuous vertices prescribed in the geometric prompt and that of the generated profile.

## 5.2 QUANTITATIVE RESULTS

Table 2 presents a comparative analysis of evaluation metrics across five models: a baseline model trained without construction sequences; a construction steps model trained with construction sequences with the same hyperparameters as the baseline model; and three RL finetuned variants of the construction steps model. These aligned variants, ReMax, GRPO, and RLOO, are optimized using reward functions defined in Section 4.2.1. All models are evaluated with greedy sampling of top k equals to one.

The introduction of construction sequences leads to substantial improvements over the baseline across all validity metrics and most of the prompt satisfaction metrics. Syntactic validity increases

from 88.1% to 94.0%, while the proportion of non-self-intersecting profiles increases from 81.9% to 84% and compliance with minimum edge length increased from 88.2% to 94.3%. Among the prompt satisfaction metrics, the most notable gains are observed in line segment adherence and mirror symmetry. These results confirm that integrating construction sequences alone significantly enhances both structural validity and alignment with geometric constraints.

Further gains are realized through RL-based alignment. The aligned variants consistently outperform the unaligned construction steps model across syntactic and geometric validity metrics, including self-intersection avoidance and minimum edge length compliance. For instance, both RLOO and GRPO achieve more than 6% reduction in generating self-intersecting geometries. Notably, although the reward functions are explicitly designed to optimize geometric validity, we observe consistent and often substantial gains across a broad set of geometric prompting metrics, including area accuracy, bounding box alignment, symmetry, and hole placement. This suggests that structural improvements induced by alignment not only satisfy low-level constraints but also enhance higher-level geometric properties, even when these are not directly incentivized during optimization.

Table 2: Comparison of key metrics among different models

| Metrics | Baseline model | Construction steps model | Construction steps model (ReMax) | Construction steps model (GRPO) | Construction steps model (RLOO) |
|---|---|---|---|---|---|
| Syntactic validity (↑) | 0.881 | 0.940 | 0.945 | 0.975 | **0.976** |
| No self-intersection (↑) | 0.819 | 0.840 | 0.853 | 0.903 | **0.905** |
| No short edges (↑) | 0.882 | 0.943 | 0.948 | 0.976 | **0.978** |
| Difference in area (↓) | 0.253 | 0.238 | 0.210 | 0.170 | **0.162** |
| Line segment distance (↓) | 0.00313 | 0.00152 | **0.00043** | 0.00082 | 0.00111 |
| Line segment ratio (↑) | 0.963 | **0.983** | 0.981 | 0.978 | 0.976 |
| Center-of-gravity distance (↓) | **0.0247** | 0.0267 | 0.0285 | 0.0254 | 0.0265 |
| Hole center distance (↓) | 0.0255 | 0.0318 | **0.0153** | 0.0232 | 0.0402 |
| Mirror IoU (↑) | 0.818 | 0.859 | 0.865 | **0.886** | **0.886** |
| Outer bounding box IoU (↑) | **0.990** | 0.984 | 0.981 | 0.985 | 0.983 |
| Tangent continuous vertices difference (↓) | 0.0907 | **0.0786** | 0.0799 | 0.0814 | 0.0816 |

## 6 QUALITATIVE RESULTS

Figure 2 presents qualitative comparisons of profiles generated by different models. The first column illustrates the geometric prompt used to drive the generation process, while the other columns show the generated profile geometry. These results demonstrate that in cases where the baseline model fails to adhere to the complex structural constraints encoded in the prompts, the construction steps model and its aligned variants can successfully produce geometrically valid generations faithful to the design specification. Further qualitative results are shown in Appendix D and E. These demonstrate how the geometric prompt can be used to control the shape of the generated profile and include examples of the full construction sequences and the family of shapes which can be obtained from them by varying the driving parameters.

## 7 CONCLUSION

In this work, we introduced a new sequence representation for CAD generation, which constructs profiles using a sequence of simple geometric construction steps. Adding these construction steps

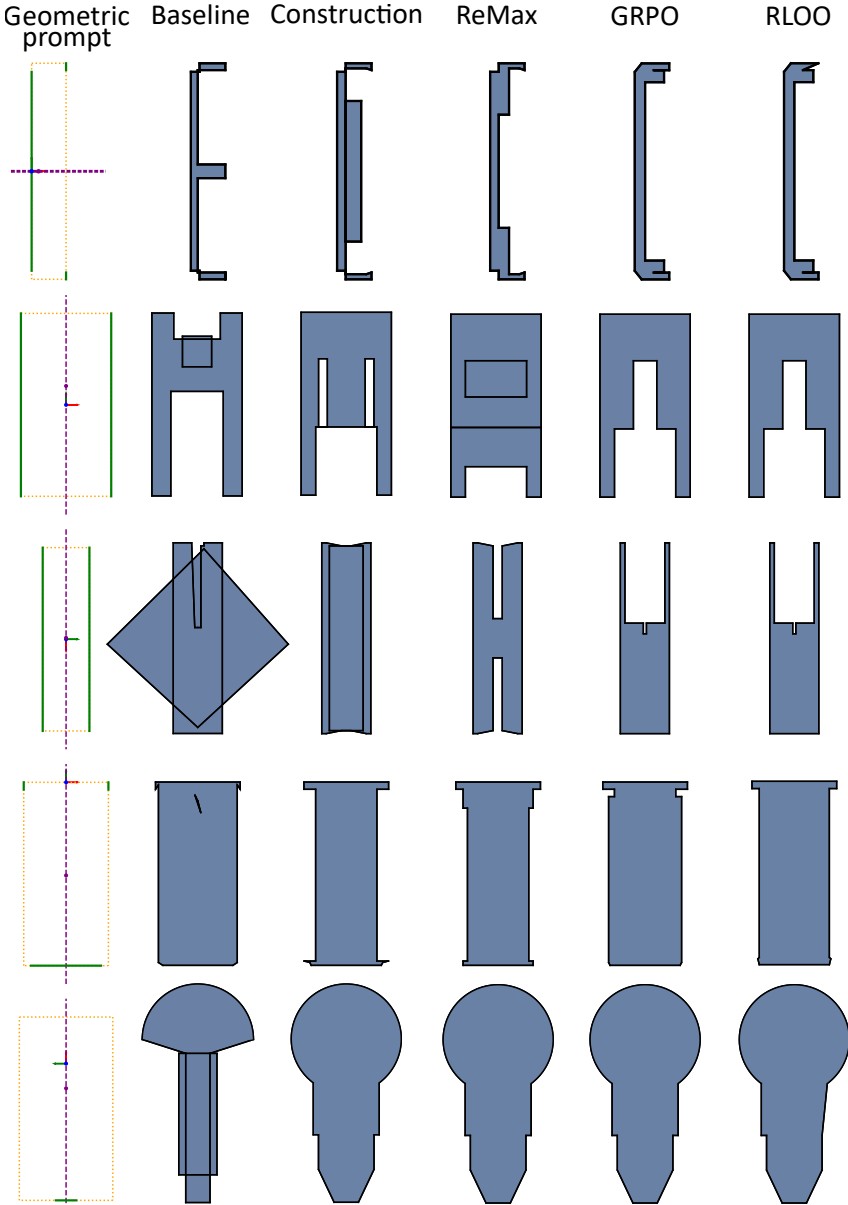

Figure 2: Visual comparison of profiles generated with the base model without construction sequences, the construction sequences model and the aligned models.

between the designers input and the final shape improves the generation quality, and promotes adherence to design requirements. Furthermore, we showed that reinforcement learning, guided by reward functions that penalize self-intersections, achieves consistent improvements across a range of metrics, including those not explicitly targeted. As the generated sequences can be replayed with floating point precision, they overcome the accuracy limitations of previous methods. Additionally, by reducing the degrees of freedom into a small set of parameters, the resulting shapes can be manipulated parametrically, much like edits in a parametric CAD system.

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
