# OpenReview forum: "Draw it like Euclid:  Teaching transformer models to generate CAD profiles using ruler and compass construction steps"
_ICLR.cc/2026/Conference — ICLR 2026 Conference Withdrawn Submission_

### Official Review · Reviewer_M4sR · 2025-10-25

**Soundness:** 2
**Presentation:** 2
**Contribution:** 2
**Rating:** 2
**Confidence:** 4

**Summary:**

This paper proposes a method that predict step-by-step CAD geometric sequences to match the input constraints described in the geometric prompt. The proposed method combines a supervised learning and a reinforcement learning steps. Overall, the results demonstrated in the paper are all in 2D instead of 3D CAD results.

**Strengths:**

- The step-by-step construction sequence prediction is an important task and challenging problem to solve. And it makes sense to predict CAD construction sequence to better fulfill the desired constraints.
- The evaluation criteria used in the experiment are reasonable and can be used for evaluating the quality.

**Weaknesses:**

- The dataset construction steps are quite confusing. It is unclear to me that what is the input and what is the output of the dataset construction steps. What is the purpose of converting B-Rep using Open Cascade? And it is quite difficult to understand the steps without any illustration, especially for the heuristic algorithms used for the extraction of the construction sequences from the raw profile geometry.
- The learning architecture and process seems just replicate what previously proposed in relevant fields and contexts. And the analogy to chain-of-thought seems just a story framing without any real relationship with the proposed method since the autoregressive models are not entirely new. This kind of sequential prediction has been proposed before and I think it is not necessarily has anything to do with chain-of-thought. I would recommend the authors to remove the claim of the relationship with chain-of-thought.
- The lack of comparison with other methods makes it difficult to judge how well the method performs and improves against other method with similar goal.

**Questions:**

- I am curious about why there is no specific comparison against other methods?
- I think the complexity of the reconstructed profiles are too low. And I think it would be better to have results that put 5 or 6 separated planes together for a complete shape if I understand correctly. I am curious about why the authors did not demonstrate such results?
- Is it necessary to have a new DSL? With the limited functionality demonstrated in the paper, I am a bit confused why to propose a new DSL? Meanwhile, I think the prediction generated by the model can be treated as many different actions directly. That means, the framing of having this DSL is totally not necessary.
- How long it takes to generate a profile? And how the computational time effected by the number of elements in the profile?

---

### Official Review · Reviewer_oHNs · 2025-10-25

**Soundness:** 2
**Presentation:** 1
**Contribution:** 2
**Rating:** 2
**Confidence:** 4

**Summary:**

This paper targets generating 2D CAD profiles, aiming to address the limitations of current CAD generation in precision and the need for postprocess constraint solving. This paper proposes a new approach to generate editable, parametric CAD profiles through sequences of simple geometric construction, including ruler, compass, and protractor operations. The key technical modules include (1)extracting construction sequences from the ABC dataset using heuristic algorithms, (2)training an autoregressive Transformer to predict full sequences (3).further refining via RL with sequence-level policy gradients methods (ReMax, GRPO, RLOO). Experiments on test data show the construction model outperforming the baseline. Additionally, using RL with reward functions which discourage self-intersecting geometry further leads to improvement over differents metrics .

**Strengths:**

The paper introduces a domain-specific language that builds 2D profile geometry as a sequence of ruler, compass and protractor construction steps.

The RL module id effective. It not only optimizes targeted validity metrics but also improves prompt satisfaction metrics, which is not explicitly targeted.

**Weaknesses:**

This paper only focuses on 2D profiles but ignores extrusion or revolution to 3D, which is very common and much more important in CAD. The qualitative results shown in the paper are neither sufficient nor complex enough. Therefore, it is not clear if this work is effective for full complex CAD workflows.

RL-based CAD command sequence generation methods have been extensively studied, like CSGNet and RLCAD

The dataset creation relies on "simple heuristic algorithms" for analysis and graph simplification, which is handcrafted. How do you ensure that these heuristics “simulate the input of a designer” as claimed? There is also no sensitivity analysis showing how heuristic choices affect sequence quality.

The geometric prompt is a crucial input, but it is treated as fixed without any ablation studies. We cannot determine whether these improvements truly come from the constructed sequences, or are merely because the prompt design is sufficiently powerful.

There are no experiments either on sequence length, profile complexity (like the number of curves), which do not give in-depth insights of the properties of CoT.

**Questions:**

Is it possible to replace flat sequences with a tree-based DSL to handle hierarchical constructions?

How is offset direction enforced in LineOffsetLine? What floating-point tolerance for intersections in replay? These omissions will affect reproducibility.

---

### Official Review · Reviewer_djqc · 2025-10-31

**Soundness:** 2
**Presentation:** 2
**Contribution:** 2
**Rating:** 4
**Confidence:** 4

**Summary:**

This paper introduces a novel approach to CAD profile generation by training transformer models on sequences of geometric construction steps (ruler, compass, and protractor operations) rather than directly generating final geometry. The key insight is that intermediate construction sequences act as a "chain of thought" that improves generation quality, similar to CoT reasoning in language models.

**Strengths:**

1- Novel and Well-Motivated Approach: The analogy to geometric construction and CoT reasoning is compelling. The idea of generating intermediate construction geometry mirrors how human designers actually work, making it both interpretable and practically motivated.

2- Strong Empirical Results: The construction sequence model shows substantial improvements over the baseline across multiple metrics

**Weaknesses:**

1- Weak Baseline Comparisons: The paper only compares against a single baseline (their own model without construction sequences).

2- Prompting bias & generality. Prompts include convex-hull lines, hole centers, symmetry lines, areas, and bounding boxes—powerful signals that may over-constrain the task. It is unclear how methods fare under noisy, partial, or incorrect promptsCan you provide quantitative comparisons to prior state-of-the-art methods (SkexGen, BRepGen, etc.) on your test set?

**Questions:**

1- Can you provide quantitative comparisons to prior state-of-the-art methods (SkexGen, BRepGen, etc.) on your test set?
2- For GRPO/RLOO vs. ReMax, how many sequences per prompt, total updates, and compute?
3- Will you release the extraction code, DSL spec, and trained checkpoints to ensure reproducibility?

---

### Official Review · Reviewer_xzBb · 2025-11-01

**Soundness:** 2
**Presentation:** 2
**Contribution:** 2
**Rating:** 4
**Confidence:** 3

**Summary:**

This paper introduces a new geometric representation format wherein an profile is generated as a series of geometric constructions. Similar to the Chain-of-Thought procedure utilized with LLM-inference, the authors hypothesize that transformer models could benefit from sequential geometric constructions building to the final profile being generated.  They provide a novel dataset of 318,000 profiles, comprised of prompt geometries and sequence constructions. The authors design a reward formulation, prioritizing avoiding self-intersection and validity, and train a transformer based model to generate CAD profiles via reinforcement learning. This paper utilizes three commonly used policy gradient approaches with their reward formulation and evaluate their approach on both prompt-satisfaction and validity metrics.

**Strengths:**

- The authors provide a strong dataset of CAD profiles with Geometric “thoughts” which could be helpful in training multimodal model capable of reasoning in a multi-dimensional output space.
- The authors validate their hypothesis on intermediate geometric constructions by showing that the inclusion of these constructions as “thoughts” improves the model’s ability to produce more syntactically valid CAD profiles, with less self-intersection.

**Weaknesses:**

- My primary issue with this paper is that the paper is missing baselines to help contextualize the performance of their proposed approach. Both GPT-4, Claude-Sonnet and Deepseek-V3 have all been leveraged to generate CAD models or CAD profiles in prior work. If the authors could not find any comparable prior works, comparing to one of these off-the-shelf models would have helped contextualize the performance of the proposed model in this paper.
    - Furthermore, integrating these geometric constructions into the prompt of a large-scale, pretrained multimodal language model (MLM) could help in understanding whether this intermediate geometric construction procedure is beneficial across models.
- It was unclear to me the format of the output data that is predicted by the model proposed in this paper. I think this paper would benefit from an additional subsection, either in their methodology or a preliminaries section, theoretically describing their modelling approach for CAD profile generation.

**Questions:**

- How did the authors arrive at their final reward construction? Did the authors conduct ablations on the reward function to understand the impact of each of the reward components?
- Are the authors planning to publicly release this dataset? As the dataset is a critical contribution of this work, I think the dataset should be publicly released if this paper is to be accepted.
- There are a few important citations which are relevant to the work being proposed, that I would recommend adding to the paper [1,2,3,4,5,6].

[1] - Guan, Yandong, et al. "CAD-Coder: Text-to-CAD Generation with Chain-of-Thought and Geometric Reward." arXiv preprint arXiv:2505.19713 (2025).
[2] - Yin, Xiaolong, et al. "Rlcad: Reinforcement learning training gym for revolution involved cad command sequence generation." arXiv preprint arXiv:2503.18549 (2025).
[3] - Yavartanoo, Mohsen, et al. "Text2cad: Text to 3d cad generation via technical drawings." arXiv preprint arXiv:2411.06206 (2024).
[4] - Alrashedy, K., Tambwekar, P., Zaidi, Z., Langwasser, M., Xu, W., & Gombolay, M. (2024). Generating cad code with vision-language models for 3d designs. arXiv preprint arXiv:2410.05340.
[5] - Xueyang Li, Yu Song, Yunzhong Lou, and Xiangdong Zhou. 2024. CAD Translator: An Effective Drive for Text to 3D Parametric Computer-Aided Design Generative Modeling. In Proceedings of the 32nd ACM International Conference on Multimedia (MM '24). Association for Computing Machinery, New York, NY, USA, 8461–8470. https://doi.org/10.1145/3664647.3681549
[6] - Govindarajan, Prashant, et al. "Cadmium: Fine-tuning code language models for text-driven sequential cad design." arXiv preprint arXiv:2507.09792 (2025).

---

### Note · Authors · 2025-11-17

**Comment:**

We would like to thank all the reviewers for their time and effort giving us this useful feedback.  We have decided to withdraw the paper in order to add comparisons with other methods.

**Withdrawal Confirmation:**

I have read and agree with the venue's withdrawal policy on behalf of myself and my co-authors.